# R-propranolol is a small molecule inhibitor of the SOX18 transcription factor in a rare vascular syndrome and hemangioma

Jeroen Overman[1†], Frank Fontaine[1†], Jill Wylie-Sears[2], Mehdi Moustaqil[3],
Lan Huang[2], Marie Meurer[4], Ivy Kim Chiang[1], Emmanuelle Lesieur[1], Jatin Patel[5],
Johannes Zuegg[1], Eddy Pasquier[4], Emma Sierecki[3], Yann Gambin[3],
Mohamed Hamdan[6], Kiarash Khosrotehrani[5], Gregor Andelfinger[7],
Joyce Bischoff[2*], Mathias Francois[1*]

[1]Institute for Molecular Bioscience, The University of Queensland, Brisbane, Australia; [2]Vascular Biology Program, Department of Surgery, Boston Children's Hospital, Harvard Medical School, Boston, United States; [3]Single Molecule Science, Lowy Cancer Research Centre, The University of New South Wales, Sydney, Australia; [4]Centre de Recherche en Cancérologie de Marseille (CRCM Marseille Cancer Research Centre), Inserm UMR1068, CNRS UMR7258, Aix-Marseille University UM105, Marseille, France; [5]Translational Research Institute, Diamantina Institute, The University of Queensland, Brisbane, Australia; [6]Dubai Healthcare City, Dubai, United Arab Emirates; [7]Department of Pediatrics, University of Montreal, Ste-Justine University Hospital Centre, Montréal, Canada

**\*For correspondence:**
joyce.bischoff@childrens.harvard.edu (JB);
m.francois@imb.uq.edu.au (MF)

[†]These authors contributed equally to this work

**Competing interests:** The authors declare that no competing interests exist.

**Abstract** Propranolol is an approved non-selective β-adrenergic blocker that is first line therapy for infantile hemangioma. Despite the clinical benefit of propranolol therapy in hemangioma, the mechanistic understanding of what drives this outcome is limited. Here, we report successful treatment of pericardial edema with propranolol in a patient with Hypotrichosis-Lymphedema-Telangiectasia and Renal (HLTRS) syndrome, caused by a mutation in *SOX18*. Using a mouse pre-clinical model of HLTRS, we show that propranolol treatment rescues its corneal neo-vascularisation phenotype. Dissection of the molecular mechanism identified the R(+)-propranolol enantiomer as a small molecule inhibitor of the SOX18 transcription factor, independent of any anti-adrenergic effect. Lastly, in a patient-derived in vitro model of infantile hemangioma and pre-clinical model of HLTRS we demonstrate the therapeutic potential of the R(+) enantiomer. Our work emphasizes the importance of SOX18 etiological role in vascular neoplasms, and suggests R(+)-propranolol repurposing to numerous indications ranging from vascular diseases to metastatic cancer.
DOI: https://doi.org/10.7554/eLife.43026.001

## Introduction

Human mutations in the *SOX18* gene give rise to Hypotrichosis-Lymphedema-Telangiectasia and Renal syndrome (HLTRS) (*Irrthum et al., 2003*; *Moalem et al., 2015*). Most disease-causing *SOX18* mutations lead to a premature truncation at the C-terminus due to an early stop codon, producing defective protein products that act in a dominant negative fashion. The mutated SOX18 protein can still bind to chromatin but fails to recruit its protein partner(s) to transactivate target genes, and thereby disrupts the endogenous function of SOX18. Further, the dominant-negative mutant prevents other SOX transcription factors to act redundantly and rescue the molecular pathway (*Hosking et al., 2009*). From birth to adolescence HLTRS patients develop symptoms such as

lymphedema and abnormalities to hair and nails for which very limited effective treatment options exist. Patients suffering from HLTRS display a certain level of variability in their phenotypic characteristics with distinct clinical manifestations (see for review *Valenzuela et al., 2018*) that likely depend on the specific type of *SOX18* mutation and how it interferes with SOX18 transcriptional regulation.

A recent report of an HLTRS patient described symptoms so mild that the initial diagnosis failed to identify this syndrome, mostly due to the absence of lymphedema up until late adolescence (*Wünnemann et al., 2016*). Eventually, the patient was subjected to whole-exome sequencing, which led to the discovery of a premature stop-codon in *SOX18*, p.Q161*; with such a mutation the patient would be expected to present a severe phenotype. A search for a potential cause of the milder course of the syndrome in this patient revealed that she had been treated with high doses propranolol since a young age due to a significant dilation of the thoracic artery causing a high blood pressure. The milder phenotype observed in this HLTRS patient coincident with the course of the propranolol treatment raised the possibility that this drug has a SOX18-dependent molecular mode of action in addition to its β-blocking activity.

The main indications for propranolol therapy are hypertension, ischemic heart disease, arrhythmia, migraine and essential tremor. Sensitivity to propranolol does not always correlate with the expression levels of the β-adrenergic receptors (*Wolter et al., 2014*; *MacDonald et al., 2006*; *Chang et al., 2015*; *Powe et al., 2010*). Propranolol use in obstructive hypertrophic cardiomyopathy revealed its beneficial effects in infantile hemangioma (*Léauté-Labrèze et al., 2008*), raising the possibility that β-adrenergic receptor signaling, and thus by inference, propranolol, modulates VEGF levels (*Chen et al., 2013*; *Ozeki et al., 2013*; *Lavine et al., 2013*). Conclusive proof of such a mode of action is still lacking. Further, this drug is a racemic mixture with R(+) and S(-) enantiomers in equal proportions. The S(-) form displays most of the anti-β-adrenergic activity, whereas the R(+) enantiomer is carried over during drug synthesis. This makes the mode of action of propranolol even more elusive since therapeutic outcomes may result from a combination of effects from the two enantiomers.

In this study, we report an additional case of HLTRS successfully treated with propranolol and describe a R(+) enantiomer-specific mode of action that acts through interference of SOX18 transcriptional activity and thereby functions as an anti-angiogenic agent whereas the S(-) enantiomer does not. This finding paves the way for new therapeutic applications using SOX18 transcription factor as a molecular target in the context of vascular diseases.

## Results

Based on the observation that a previous case of HLTRS may have benefited from a long-term exposure to propranolol (*Wünnemann et al., 2016*), we considered propranolol therapy for another HLTRS patient (*Bastaki et al., 2016*) in an attempt to mitigate the severity of his symptoms. The patient was diagnosed with HLTRS at 11 months of age, after presenting with alopecia, edema and telangiectasia (*Bastaki et al., 2016*). Exome sequencing revealed that this patient had a truncation mutation in *SOX18* (SOX18 c.492_505dup). In September 2015, at 17 months of age, he developed progressive pericardial effusion over 3 weeks, which required pericardiocentesis. Fluid analysis showed transudate. Two weeks after the procedure, pericardial effusion re-accumulated progressively over 6 weeks, despite strong anti-inflammatory regimens including 3 weeks of oral ibuprofen and 10 days of oral prednisone. Past clinical experience in HLTRS suggested the progression of this pericardial effusion was intractable. The effusion was large enough to cause hemodynamic compromise and to necessitate another pericardiocentesis. After obtaining parental consent, oral propranolol was introduced and increased gradually every 2–3 weeks from 0.8 mg/kg/day in three divided doses to reach a maximum of 4.1 mg/kg/day, as tolerated clinically (pulmonary wheezing). We observed spontaneous and rapid resolution of the pericardial effusion after reaching a dose threshold of 3 mg/kg/day. Bronchial side effects limited the maximal dosing in this patient to 4 mg/kg/day (*Figure 1A and B*). The positive outcome in this second HLTRS patient harboring SOX18 mutation suggested propranolol might act via an alternative pathway than the well characterized β-adrenergic receptor blockade.

We next investigated the potential of propranolol to exert β-adrenergic independent effects in cancer cell lines and in human endothelial cells (*Figure 1C–D*). Survival of primary placentally-derived endothelial colony forming cells (ECFC) was measured in presence of increasing concentrations (5–

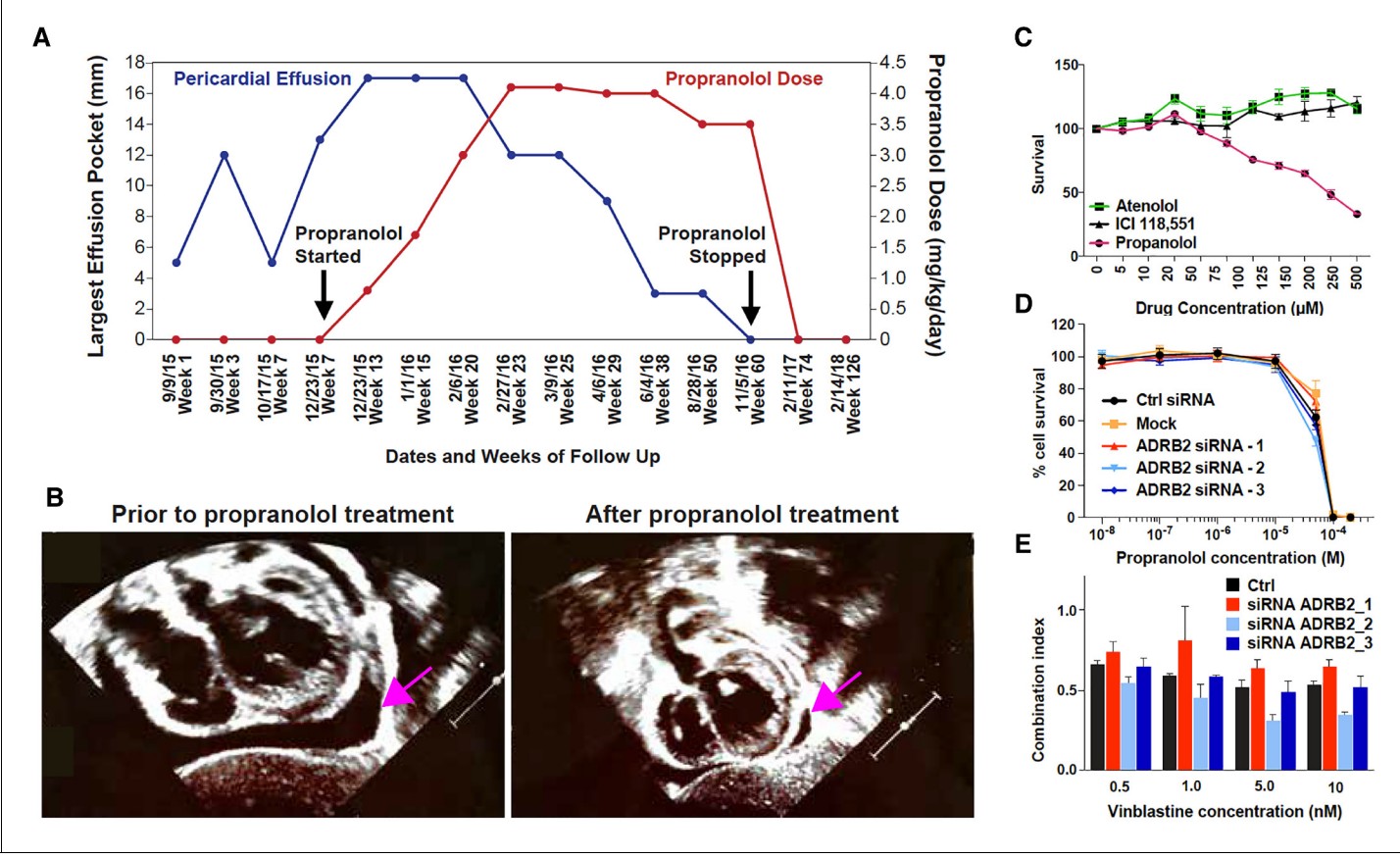

**Figure 1.** Propranolol treatment alleviates pericardial effusion severity in HLTRS patient and mediates β-adrenergic independent effects. (**A**) A 17 months old HLTRS patient was treated with propranolol, starting at 0.8 mg/kg/day in three divided doses and increasing gradually every 2–3 weeks to maximum of 4.1 mg/kg/day (red curve). In parallel the volume of ventricular peri-cardial effusion was measured at the end of the diastole (blue curve). (**B**) Echocardiography revealed that pericardiocentesis was not required anymore after propranolol treatment due to significant reduction in pericardial effusion (pink arrow) which did not recur as of June 2018 (time of the study). (**C**) Fetal endothelial colony forming cells (ECFC) were isolated from term placenta from healthy donors, expanded for three passages, and subjected to propranolol treatment followed by analysis of survival (percentage) as compared to vehicle control (DMSO). Propranolol affected the survival of ECFC at equivalent doses whereas Atenolol (specific β1 blocker) and ICI118,551 (specific β2) did not. (**D**) Cell survival assay performed on ISO-HAS angiosarcoma cells after transfection with three different siRNA sequences targeting *ADRB2* and following 72 hr incubation with propranolol (racemic mixture). Alamar Blue assay ; *Points*, mean of at least four independent experiments ; *Error bars*, standard error. (**E**) Combination indexes of propranolol and vinblastine in ISO-HAS angiosarcoma cells after transfection with three different siRNA sequences targeting *ADRB2* and following 72 hr drug incubation (50uM). Alamar Blue assay ; *Bars*, mean of at least four independent experiments ; *Error bars*, standard error. Statistical analysis for C was performed using Mann-Whitney non parametric t-test and for D-E using an unpaired two-tailed t test.

DOI: https://doi.org/10.7554/eLife.43026.002

The following figure supplement is available for figure 1:

**Figure supplement 1.** β-adrenergic independent effects of propranolol.
DOI: https://doi.org/10.7554/eLife.43026.003

500 µM) of propranolol and specific β1 (Atenolol) and β2 (ICI 118,551) adrenergic-receptor blockers. Results showed that pharmacological inhibition of specific β-adrenergic receptors did not replicate the marked effect propranolol had in the ECFC survival assay (*Figure 1C*).

To further explore β-adrenergic independent effects of propranolol we knocked down *ADRB2* in a propranolol-sensitive angiosarcoma cell line (ISO-HAS, *Figure 1—figure supplement 1A*). The cytotoxic activity of propranolol (*Figure 1D*) and its enhancement of vinblastine cytotoxicity (*Figure 1E*) were unaltered when *ADRB2* was silenced. These results show that the anti-survival and sensitization to chemotherapy activities of propranolol are independent of β-adrenergic receptor expression. Of note, *ADRB1* is expressed at low levels in the angiosarcoma cell line (not shown) but

is expressed at higher levels in the SHEP neuroblastoma cell line, where its expression can be reduced by siRNA (*Figure 1—figure supplement 1B*). Similar results were obtained when using a non-endothelial cell type (SHEP neuroblastoma cell line) whereby knock-down of *ADRB1* and *ADRB2* alone or in combination did not affect the ability of propranolol to block the proliferative response and its ability to enhance response to chemo-therapies (*Figure 1—figure supplement 1C and 1D*).

The observation that adrenergic receptors are not required for the cellular response to propranolol combined with the clinical benefit of propranolol treatment in the context of HLTRS suggests that this FDA approved drug may act by directly modulating *SOX18* protein activity .

Although SOX18 is involved in a variety of patho-physiological processes (*Olbromski et al., 2018*), HLTRS is a rare and severe vascular disorder. In order to further investigate propranolol in this disease scenario, we used a mouse model of HLTRS, SOX18 *Ragged Opossum* (*RaOp*) (*Pennisi et al., 2000*; *Slee, 1957*). This mouse mutant is considered a murine counterpart of human HLTRS syndrome since it displays the same range of cardio-vascular and hair follicle defects and exhibits the same type of dominant-negative mutation in the *Sox18* gene (*Pennisi et al., 2000*; *Downes et al., 2009*; *François et al., 2008*; *Villani et al., 2017*). The *Ragged* mouse is characterized by pathologic corneal neo-vascularisation (CNV, *Figure 2—figure supplement 1A-C*) (*François and Ramchandran, 2012*). This phenotypic hallmark offers an indirect biological readout in vivo for aberrant SOX18 function in the control of vessel outgrowth. Propranolol was tested on CNV in the *RaOp* mouse model to validate its therapeutic potential. Because of early postnatal onset of HLTRS defects (between postnatal day (P)8–22), oral treatment with propranolol (25 mg/kg/day), or vehicle PBS, was initiated at P8 and continued daily through to P28. Both wild type and mutant littermates were subjected to identical treatment schemes in order to safeguard consistent maternal feeding and grooming.

Treatment with propranolol (21 days) did not affect the gross morphological appearance of either wild type or *RaOp* pups, nor did it significantly affect the weight of the *RaOp* animals (*Figure 1—figure supplement 2A and B*, n = 5–10 mice per group). We did not observe any obvious side effects, consistent with the safety profile of propranolol established over the past decades.

Propranolol treatment of *RaOp* mice, however, led to an almost complete rescue of the CNV phenotype (*Figure 2A*). The corneas from propranolol-treated mutant animals were devoid of blood vessels (*Figure 2A*, Endomucin- and ERG-positive cells) in five out of five mice, and one mutant pup only had a mild penetration of blood vessels into the cornea (*Figure 2A*). These pre-clinical data indicate that propranolol treatment is an effective therapeutic strategy to block aberrant vessel growth in a mouse model of HLTRS and is indicative of on-target engagement on SOX18 protein in vivo.

These in vivo data prompted us to assess the transcriptional activity of Sox18 in presence of propranolol. *Sox18* was overexpressed in COS-7 cells, along with a synthetic construct containing a luciferase reporter gene fused to the Vcam1 promoter fragment, a direct target of Sox18 (*Hosking et al., 2004*). In this assay, we interfere with wild type Sox18 function using the *RaOp* dominant-negative protein. We found that *RaOp* completely abolished the activity of the wild type protein at ratios as low as 1:30 *RaOp:Sox18* WT (*Figure 2B*). We next tested whether propranolol could rescue the dominant-negative inhibitory effect of *RaOp* reproduced in this cell-based system. Propranolol restored *SOX18* wild type functionality in a dose-dependent manner; it significantly increased Vcam1 reporter promoter activity to base line levels at 15 µM (p-value≤0.001) (*Figure 2B*).

Lastly, to demonstrate the molecular mode of action of propranolol on SOX18 protein partner recruitment, we used an ALPHAScreen assay to measure pairwise protein-protein interaction (PPI). SOX18 homodimer and SOX18/RaOp (Q161*) heterodimer formation was measured in absence or presence of propranolol (*Figure 2C*). Propranolol showed a mild efficacy at disrupting SOX18 homodimer and was able to interfere more effectively with the assembly of the non functional SOX18/RaOp protein complex. An effector of the NOTCH signaling pathway (RBPJ) was recently shown to act as a SOX18 protein partner (*Overman et al., 2017*; *Fontaine et al., 2017*). In addition to disrupting SOX18 homodimer assembly (*Figure 2C*), we found that propranolol inhibits the association of SOX18 with RBPJ at low micro-molar range (5 µM) (*Figure 2—figure supplement 2C*). Taken together these in vivo pre-clinical data and in vitro results strongly suggest that one mechanism by which propranolol mediates its anti-angiogenic effects is via direct interference with a *SOX18*-dependent transcriptional activation.

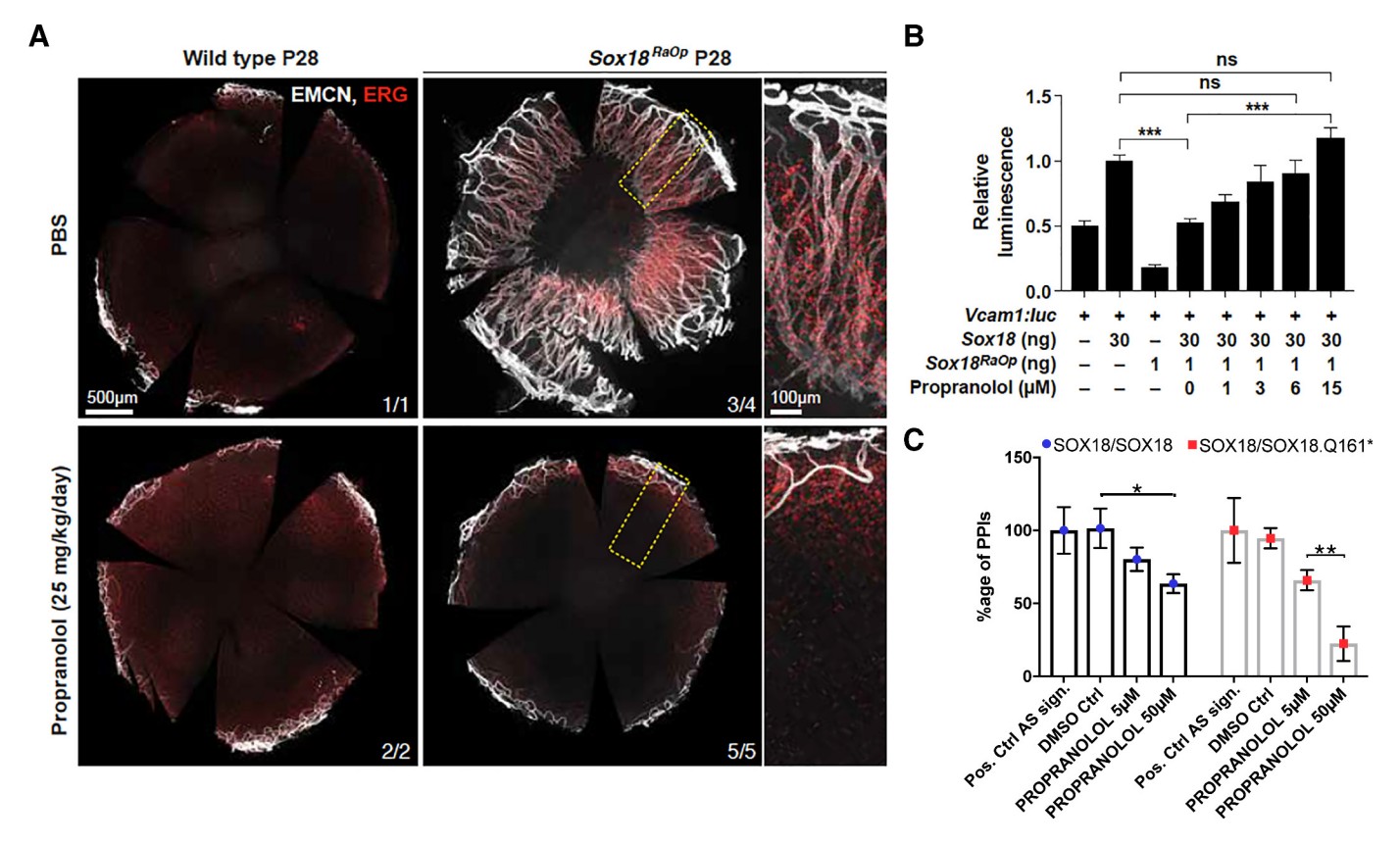

**Figure 2.** Propranolol rescues corneal neo-vascularization phenotype in a mouse pre-clinical model of HLTRS and SOX18 dominant-negative transcriptional repression via protein-protein interaction disruption. (**A**) Fluorescent images of corneal flat mounts, showing blood vessel penetration into the cornea at P28 stage using endothelial cell markers ERG and endomucin (EMCN). *Sox18* WT and *RaOp* mice were treated from P8 to P28 with either vehicle PBS or propranolol. Propranolol has no obvious effect on WT cornea, but prevents CNV in *RaOp* pups. Number of predominant phenotype shown in bottom right. Scale bar left 500 μm, right 100 μm. (**B**) COS-7 cells were transfected with SOX18 responsive Vcam1:*luciferase* construct and a combination of *Sox18* wild type plasmid DNA and RaOp plasmid DNA. *RaOp* behaves in a dominant negative fashion and is capable to inhibit SOX18 WT function even at low 30:1 (w/w DNA) allelic ratios. Addition of propranolol to the media rescues SOX18 dependent activity of the Vcam1 promoter in presence of *RaOp*. Effect is concentration dependent and normal SOX18 activity on this construct is achieved at 15 μM propranolol. *Sox18*. *** p-value≤0.001, Kruskal-Wallis multiple comparison test. Data shown is mean ± SD of n ≥ 8. (**C**) The bar graph shows ALPHAScreen signal as a measure of the level of protein-protein interaction between SOX18 and its mutant counterpart RaOp (red square) and SOX18 homodimer formation (blue dot) in absence or presence of propranolol treatment. Propranolol is a small compound with the ability to disrupt SOX18 self-recruitment. Statistical analysis in 2B one-way ANOVA with Bonferroni post-hoc test and in 2C ANOVA Sidak's multiple comparison test. Analysis of the protein pair by ALPHAScreen assay was performed in three different biological experiment with three technical replicates.

DOI: https://doi.org/10.7554/eLife.43026.004

The following figure supplements are available for figure 2:

**Figure supplement 1.** Time course of the corneal phenotype.
DOI: https://doi.org/10.7554/eLife.43026.005
**Figure supplement 2.** Effects of propranolol on the overall morphology of treated *RaOp* mice and disruption of SOX18/RBPJ protein-protein interaction.
DOI: https://doi.org/10.7554/eLife.43026.006

We have previously identified, developed and validated a small compound, Sm4, that disrupts SOX18 homodimer formation in turn affecting SOX18 ability to recruit RBPJ. Further, we have shown potential therapeutic applications in managing vascular over-growth in pre-clinical model of breast cancer metastasis (*Overman et al., 2017*). When using Sm4 as a reference, propranolol was equally effective at disrupting SOX18/RBPJ PPI however not as effective at blocking SOX18 homodimer formation (*Figure 2—figure supplement 2C*). Altogether, these results suggest that the same molecular mode of action of propranolol accounts for both the rescue of dominant-negative SOX18 loss of

function by disruption of the non-functional SOX18/RaOp (Q161*) complex, and the disruption of functional SOX18 wild type PPIs, namely SOX18/RBPJ and SOX18/SOX18 dimers. In the context of the Ragged heterozygous mutation dominant negative mutation, the rescue process corresponds to the inhibition by propranolol of an inhibitor (RaOp) which in turn results to the partial restoration of the SOX18 wild type active form.

This novel mode of action of propranolol, independent of its anti-adrenergic effects, prompted us to examine its mechanism of action in infantile hemangiomas, the most common vascular tumor in infancy. Although the molecular mode of action of propranolol in infantile hemangioma therapy is largely unknown, it is the mainstay of treatment for infantile hemangioma. (*Léauté-Labrèze et al., 2008*; *Léauté-Labrèze et al., 2015a*; *Léauté-Labrèze et al., 2015b*) A variety of propranolol-induced cellular effects have been observed in infantile hemangioma-derived stem cells (HemSCs), hemangioma endothelial cells (HemECs) and hemangioma pericytes (*Lamy et al., 2010*; *Lee et al., 2014*; *Tu et al., 2013*; *Wong et al., 2012*). To learn if SOX18 is involved, we first explored the expression of SOX18 in vivo in hemangioma tissue sections from three patients and in vitro in different cell populations derived from infantile hemangiomas: *SOX18 protein was observed in a sup-population of CD31-positive endothelial cells* (*Figure 3A*, arrows) and *SOX18* mRNA was expressed in ECFC (positive control) and in HemEC, whereas HemSC showed low expression and hemangioma pericytes were completely devoid of *SOX18* transcripts (*Figure 3—figure supplement 1*).

To interrogate whether propranolol inhibits *SOX18* in this setting (*Figure 3B*), we induced patient derived HemSCs to differentiate into HemECs using VEGF-B and low serum media (*Khan et al., 2008*). *CDH5* (a pan-endothelial marker), *SOX18*, and *ADBR2* mRNAs were increased in VEGF-B-treated HemSC (*Figure 3C*) further showing a process by which differentiation of stem cells results in endothelial cells expressing SOX18. This was extended to other genes such as *CD31*, *NOTCH1*, *PLXND1* and *VEGFR1* (*Table 1* and *Figure 3—source data 1*), each of which are expressed in infantile hemangioma endothelium in vivo (*Boscolo et al., 2011*; *Nakayama et al., 2015*; *Wu et al., 2010*), further confirming the successful differentiation of HemSCs into HemECs (*Figure 3D* and *Figure 3—figure supplement 2A*). Using this differentiation assay as a read-out, we next evaluated the capacity of propranolol to inhibit the differentiation of HemSCs isolated from four infantile hemangioma patients. The pharmacologic treatments of HemSC included: propranolol, Sm4 (SOX18 small molecule inhibitor), aspirin (a negative control [*Overman et al., 2017*; *Fontaine et al., 2017*]) compared to vehicle control. Of note, SOX18 has been shown to directly up-regulate *Notch1* transcription via an intronic enhancer during arterial specification in vertebrates (*Chiang et al., 2017*), hence transcriptional output of this target gene indirectly reflects *SOX18* activity. Propranolol significantly decreased VEGF-B induced expression of all five endothelial markers in HemSC to levels also achieved by Sm4 as compared to controls (see *p*-values in *Figure 3—source data 1*).

Propranolol exists as a 1:1 mixture of the S(-) and R(+) enantiomers. The S(-) form displays potent anti-β-adrenergic activity whereas the R(+) enantiomer presents strongly reduced antagonist activity towards β2 adrenergic receptors on peripheral arteries (*Stoschitzky et al., 1995*). To test whether HemSC were affected by propranolol in a stereoselective manner, we assessed VEGF-B-induced HemSC to HemEC differentiation from two infantile hemangioma in the presence of separated enantiomers.

The R(+) enantiomer recapitulated propranolol and Sm4-induced inhibition on all markers, whereas the S(-) enantiomer showed only weak or no inhibition on hemangioma markers (*Figure 3D* and *Figure 3—source data 1*). These results are consistent with a SOX18-dependent, β-adrenergic receptor independent blockade of hemangioma endothelial differentiation, which strongly suggests that blockade of SOX18 activity is the main mode of action of propranolol in this setting. Further, analysis of VEGF-R2 expression and phosphorylation levels in ECFC and HemSC to EC differentiation assay ruled out that the propranolol racemic mixture, the R(+) enantiomer or Sm4 interfere with the VEGF signaling pathway (*Figure 3—figure supplement 2B-D*).

As a final demonstration of the therapeutic utility of the R(+) enantiomer we performed a corneal neo-vascularisation rescue assay using the *Ragged Opossum* mutant mouse model. In this in vivo setting both propranolol and the R(+) enantiomer prevented aberrant vessel growth in the corneal tissue of the heterozygous mutant animals. Of note the S(-) enantiomer was also active at disrupting corneal vessel outgrowth, suggesting that conformational changes that occur in the SOX18 mutated protein may attenuate the discrepancy in enantiomeric activities previously observed in the hemangioma endothelial differentiation model (*Figure 3—figure supplement 3*).

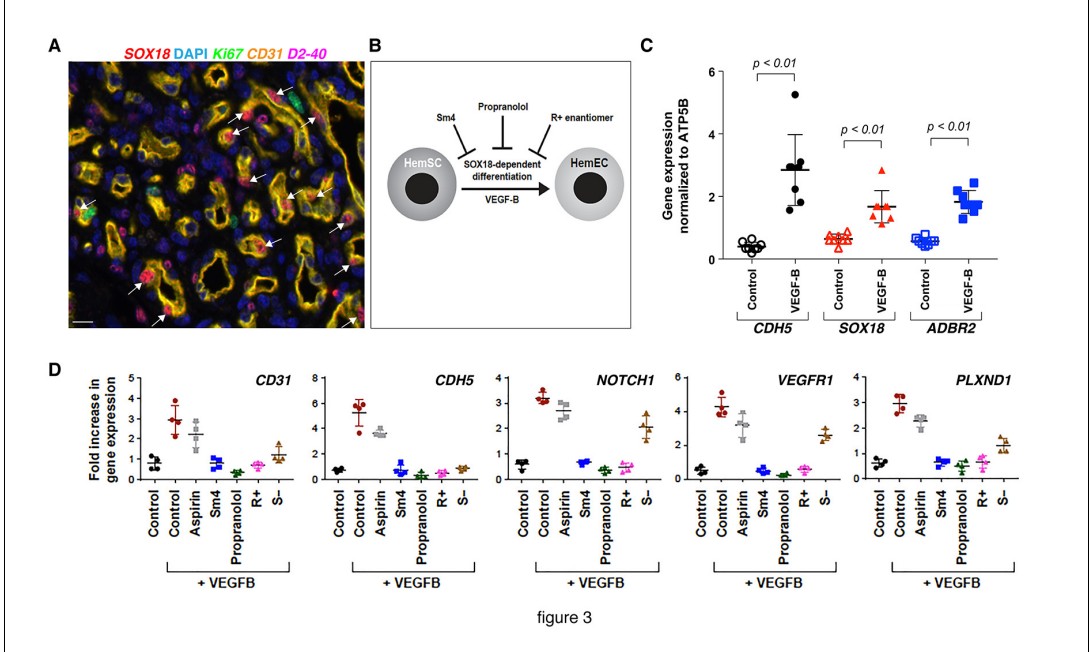

figure 3

**Figure 3.** The R(+) enantiomer of propranolol and SOX18 small molecule inhibitor halt infantile hemangioma stem cell differentiation. (A) Infantile) hemangioma tissue section stained for *SOX18* (red), *Ki67* (green), *CD31* (orange), *D2-40* (pink) and DAPI (blue) reveals the presence of *SOX18* expression in a large subset of hemangioma endothelial cells (arrows). (B) Schematic representation of infantile hemangioma stem cell (HemSC) endothelial differentiation assay. VEGF-B stimulates HemSC to differentiate into hemangioma endothelial cells (HemEC). This differentiation process is inhibited by propranolol, the R(+) enantiomer of propranolol, and by *SOX18* small molecule inhibitor Sm4 (all at 5 uM). (C) VEGF-B treatment of HemSC from four different infantile hemangiomas resulted in increased *CDH5* (an endothelial cell marker), SOX18 and ADBR2 (β2 adrenergic receptor) mRNA. Means and standard deviations are shown. (D) The effects of SOX18 inhibitor Sm4, its scaffold aspirin as a negative control, propranolol and its purified R(+) and S(-) enantiomers on HemSC-to-HemEC differentiation from two infantile hemangioma patients. Endothelial differentiation markers, *CD31* and *CDH5* and hemangioma endothelial markers *NOTCH1*, *PLXND1* and *VEGFR1* under each treatment condition in four biological replicates, determined by qPCR, were standardized as described (*Willems et al., 2008*). Means and standard deviations are shown. Statistical analysis in 3C and 3D was done using one-way ANOVA, Fisher Tests, and two-tailed two independent sample T-Tests.

DOI: https://doi.org/10.7554/eLife.43026.007

The following source data and figure supplements are available for figure 3:

**Source data 1.** Supplemental table for *Figure 3D* (p-values).

DOI: https://doi.org/10.7554/eLife.43026.012

**Figure supplement 1.** SOX18 expression in infantile hemangioma-derived cells.

DOI: https://doi.org/10.7554/eLife.43026.008

**Figure supplement 2.** SOX18 inhibitor (Sm4) and propranolol block HemSC to EC differentiation.

DOI: https://doi.org/10.7554/eLife.43026.009

**Figure supplement 3.** The racemic mixture Propranolol and its enantiomer (R(+) and S(-)) rescues corneal neo-vascularization phenotype in a mouse pre-clinical model of HLTRS.

DOI: https://doi.org/10.7554/eLife.43026.010

**Figure supplement 4.** Propranolol and its R(+) and S(-) enantiomers do not rescue the dermal lymphatic vascular phenotype of the *Ragged Opossum* mutant mouse model.

DOI: https://doi.org/10.7554/eLife.43026.011

In order to investigate whether propranolol treatment might improve the morphogenesis defects of other vascular beds, we analysed the patterning of the lymphatic vasculature in the ear skin tissue of the adult *RaOp* animals. As previously reported the *RaOp* mouse presented with a highly disorganized lymphatic vascular network (*François et al., 2008*) (*Figure 3—figure supplement 4*, top graph). In the skin, neither the propranolol racemic mixture nor its enantiomers rescued the lymphatic vascular phenotype caused by the *RaOp* mutation (*Figure 3—figure supplement 4*); this is most likely due to the post-natal timing of the treatment: lymphatic vascular defects are acquired in

the *RaOp* mutant during the initial steps of embryonic lymphangiogenesis (early to mid-organogenesis), which precedes the drug treatment window.

In summary, we suggest the R(+) enantiomer of propranolol is a potent selective inhibitor of SOX18 activity and it is sufficient to achieve the effects of propranolol racemic mixture in the context of infantile hemangioma and other SOX18-dependent vascular diseases.

## Discussion

Despite being the front line therapy for infantile hemangioma, approximately 20% of the patient population fails to respond to propranolol treatment and/or experience tumor regrowth when the medication is discontinued too early (*Bagazgoitia et al., 2011*; *Xiao et al., 2013*). Our finding opens up new treatment options using enantiopure R(+) propranolol instead of the racemic mixture. This could potentially allow a dose adjustment to achieve optimal benefit for the patients while mitigating adverse events that may result from long-term propranolol treatment, particularly in the refractory population of patients that currently do not respond to the highest dose of propranolol. In addition, the side effects associated with long-term exposure to propranolol could be greatly reduced if β-adrenergic receptors blockage would be minimized.

This discovery is of broad interest because propranolol, comprised of 2 enantiomers, has been used clinically for decades. Clinical trials of the R(+) versus the S(-) form have been performed and deemed safe with full toxicology data available (*Stensrud and Sjaastad, 1976*). Pharmacokinetic studies suggest that propranolol therapy results in an accumulation of the S(-) enantiomer following drug metabolism, which could be problematic if one is interested in the activity of the R(+) form (*Mehvar and Brocks, 2001*). Indeed toxicology studies further suggest that propranolol in its racemic form is more toxic than either enantiomer alone (*Stoschitzky et al., 1992*). This study presents indirect evidence that in a HLTRS patient treated with propranolol, SOX18 wild type function is at least partly restored. The observation that the S(-) enantiomer also has the capability to rescue the corneal neo-vascularisation phenotype in a pre-clinical model of HTLRS suggests that the effects of propranolol as a racemic mixture are likely to be a combination of the blockade of beta-adrenergic receptor activity and SOX18 RaOp mutant protein inhibition. Of importance, in the hemangioma HemSC-EC differentiation assay the R(+) enantiomer was as efficient as propranolol and more efficient than the S (-) enantiomer at blocking this differentiation process. This may suggest that the SOX18 wild type protein blockade displays some enantio-selectivity, whereas conformational changes in the SOX18/RaOp complex caused by the RaOp mutation sensitize the dominant-negative protein complex to both R(+) and S(-) enantiomers.

The studies in human patients with HLTRS and the murine *RaOP* model suggest that there may be critical time windows in which treatment with R(+) propranolol would be most efficient as an anti-angiogenic compound via a blockade of SOX18 transcriptional activity.

The low number of patients with HLTRS – less than 10 documented survivors have been reported – makes it difficult to establish genotype-phenotype relationships in greater detail. In contrast, the unambiguous phenotypic rescue by propranolol racemic mixture in two human patients and by the R(+) enantiomer in the *Ragged Opossum* mouse model is strongly supported by our in vitro mechanistic results. Our findings outline that the R(+) enantiomer is a SOX18 small molecule inhibitor with weak anti-beta adrenergic activity whereas the S(-) molecule is both a beta-blocker and a weak SOX18 inhibitor which introduces variability in net efficacy of propranolol treatment.

The strong monogenic effect of human SOX18 mutations and their rescue with propranolol allowed us to identify its mode of action. We suggest the dosing and duration of propranolol therapy may be adjusted by using R(+) propranolol in potential future trials of HLTRS and infantile hemangioma.

## Materials and methods

### Hemangioma cell isolation and culture

Specimens of infantile hemangioma were obtained under an IRB-protocol approved by the Committee on Clinical Investigation, Boston Children's Hospital. The clinical diagnosis was confirmed in the Department of Pathology. Informed consent was obtained for the specimens, according to the

Declaration of Helsinki. Single cell suspensions were prepared from the proliferating phase hemangioma specimens and HemSCs were selected and expanded as described (Khan et al., 2008; JCI).

HemSC were seeded on fibronectin (10 ng/cm$^2$, EMD Millipore, Billerica, MA)-coated plates at a density of 20,000 cells/cm2. The culture medium was composed of Endothelial Growth Medium-2 (EGM-2), which includes Endothelial Basal Medium (EBM2, Lonza, Allendale, NJ) supplemented to 10% FBS, Endothelial Growth Media-2 SingleQuots (Lonza), and 1X GPS (Mediatech Inc, Manassas, VA, 100 U/milliliter (mL) penicillin, 100 µg/mL streptomycin, 292 µg/mL Glutamine). Cells were cultured at 37°C in a humidified incubator with 5% CO$_2$.

To induce HemSC to undergo endothelial differentiation, HemSCs were seeded on fibronectin-coated plates at a density of 20,000 cells/cm$^2$ in EBM2/10%FBS. After 18–24 hr, the medium was replaced with serum-free EBM-2 containing 10 ng/ml VEGF-B (R&D Systems), 1 × insulin transferrin-selenium, 1:5000 linoleic acid–albumin, 1 µM dexamethasone, 60 µM ascorbic acid–2–phosphate. Cells were cultured in the VEGF-B, serum-free media,±indicated drugs for 5–8 days. ((±)-Propranolol hydrochloride (Sigma-Aldrich Cat# P0884), (R)-(+)-Propranolol hydrochloride (Sigma-Aldrich Cat# P0689), and (S)-(−)-Propranolol hydrochloride (Sigma-Aldrich) Cat# P8688).

Total cellular RNA was extracted from HemSCs with an RNeasy Micro extraction kit (Qiagen, Valencia, CA, #74004). Reverse transcriptase reactions were performed using an iScript cDNA Synthesis Kit (Bio-Rad, CA, USA #170–8890). qPCR was performed using Kapa Sybr Fast ABI Prism 2x qPCR Master Mix (KAPA BioSystems, MA, USA # KK4604). Amplification was carried out in an ABI 7500 (Applied Biosystems, Foster City, CA). A standard curve for each gene was generated to determine amplification efficiency. ATP5B was used as housekeeping gene expression reference. Fold increases in gene expression were calculated according to two delta C$_T$ method, with each amplification reaction performed in triplicate.

## Angiosarcoma cell culture

ISO-HAS angiosarcoma cells were kindly provided by Prof Masuzawa [Masuzawa, Int J Cancer 1999] and SHEP neuroblastoma cells were obtained from the Children's Cancer Institute Australia (Sydney, Australia). Both cell lines were grown in DMEM supplemented with 10% FCS, 2 mM L-glutamine and 1% penicillin streptomycin, and kept in culture at 37°C in a humidified atmosphere containing 5% CO2. Cell lines were regularly screened and are free from mycoplasma contamination.

## Gene silencing

ISO-HAS and SHEP cells were transfected with Lipofectamine RNAiMax (Life Technologies) and 5 nM Silencer Select siRNA sequences targeting ADRB1 and/or ADRB2 (Ambion, Life Technologies). A non-silencing control siRNA, which has no sequence homology to any known human gene sequence, was used as a negative control.

## Quantitative RT-PCR

The expression of adrenergic receptor genes ADRB1 and ADRB2 was examined in angiosarcoma and neuroblastoma cell lines following siRNA transfection using real-time quantitative RT-PCR. Total RNA was extracted using the Qiagen Mini RNeasy kit (Qiagen, Courtaboeuf, France) and RNA concentration was determined from the absorbance at 260 nm. cDNA synthesis was performed using the Quantitect Reverse Transcription kit (Qiagen). Real time PCR was run on a LightCycler 480 (Roche, Boulogne-Billancourt, France) for ADRB1 and ADRB2 using DNA primer sequences previously described (Cao et al., 2010) and endogenous control gene YWHAZ. Gene expression levels were determined using the $\Delta\Delta C_t$ method, normalized to the YWHAZ control gene.

## Cell viability assay

Cell viability assays on angiosarcoma and neuroblastoma cell lines were performed as previously described (Pasquier et al., 2013). After 72 hr drug incubation, metabolic activity was detected by addition of Alamar blue and spectrophotometric analysis. Cell viability was determined and expressed as a percentage of untreated control cells. The determination of EC50 values was performed using GraphPad Prism software (GraphPad Sofware Inc, La Jolla, CA, USA). Combination indexes (CI) were calculated for all tested drug concentrations according to the Chou and Talalay method (Chou and Talalay, 1984).

**Table 1.** qPCR primer sequences.

| Gene | Forward primer | Reverse primer |
| --- | --- | --- |
| CD31 | CACCTGGCCCAGGAGTTTC | AGTACACAGCCTTGTTGCCATGT |
| CDH5 | CCTTGGGTCCTGAAGTGACCT | AGGGCCTTGCCTTCTGCAA |
| PLXND1 | CAAGTTTGAGCAGGTGGTGGCTTT | ATTTCCCAGTCTGAGTCACAGGCA |
| NOTCH1 | CGGTGAGACCTGCCTGAATG | GCATTGTCCAGGGGTGTCAG |
| VEGFR1 | CTCAAGCAAACCACACTGGC | CGAGCTCCCTTCCTTCAGTC |
| SOX18_2 | GTGTGGGCAAAGGACGAG | AGCTCCTTCCACGCTTTG |
| ADBR2 | CACCAACTACTTCATCACTTCAC | GACACAATCCACACCATCAG |
| ATP5B | CCACTACCAAGAAGGGATCTATCA | GGGCAGGGTCAGTCAAGTC |

DOI: https://doi.org/10.7554/eLife.43026.013

Statistics qPCR data from different HemSC-to EC differentiation were standardized as described Willems (51). Data were analyzed by one-way ANOVA, Fisher Tests, and two-tailed two independent sample T-Tests. Statistical programs were from Excel and XLStat Pro.

## CNV model

All procedures involving mice were approved by the University of Queensland Animal Ethics Committee. Heterozygous RaOp pups were produced by crossing DBA/2JArc wild type females (purchased from the Animal Resource Centre) with a B6D2-RaOp/J heterozygous male (purchased from the Jackson Laboratory).

For the treatment of with propranolol, entire litters were exposed to identical treatment conditions, being either Propranolol or PBS vehicle and data shown is from three different litters per condition. Propranolol was dissolved in PBS (20 mg/ml) and orally administered at 25 mg/kg/day (l μl per gram of bodyweight per day) starting at postnatal day eight through to P28. Pups were sacrificed ($CO_2$) 2 hr after their last dose on P28, blood plasma was collected through cardiac puncture and the pictures were captured for gross morphological analysis.

Eyes were harvested, corneas were dissected and tissues were fixed in 4% PFA O/N at 4°C for morphological analysis. Fixed tissues were washed in PBTX and further dissected for gross morphological analysis and processed for immunofluorescent staining. Antibodies used were anti-mouse Endomucin (Santa Cruz Biotechnology, cat# sc-53941) anti-mouse ERG (Abcam, cat# ab92513), anti-mouse NRP2 (R and D systems, cat# AF567). Whole corneas were flatmounted on glass slides in 70% glycerol and high-resolution images were captured using a 10x (whole cornea) and 20x (detail image) objective on a Zeiss LSM 710 confocal microscope.

## Luciferase reporter assays

COS-7 cells were cultured at 37°C, 5% $CO_2$ in DMEM (Life technologies, 11995) with added FBS, sodium pyruvate, L-glutamine, penicillin, streptomycin, non-essential amino acids and HEPES (**N**-2-hydroxyethylpiperazine-**N'**−2-ethanesulfonic acid). COS-7 cells were grown in 96-well plates to 80–90% confluency, and transfected using X-tremegene 9 DNA transfection reagent (Roche, 06365787001) according to the manufacturer's instructions. Constructs used were mouse pSG5 Sox18 (30 ng), mouse pSG5 myc-RaOp (30 ng) and pGL2 Vcam1:luc (40 ng). For titration of RaOp dominant negative allelic ratios, pSG5 Sox18 was kept at 30 ng, while the amount of pSG5 myc-RaOp plasmid was reduced accordingly (supplemented to 30 ng plasmid DNA with pSG5 empty vector). Propranolol hydrochloride was added to low serum COS-7 media to obtain concentrations of 1, 3, 6, and 15 μM. Propranolol treatment extended 24 hr after the end of transfection until cells were harvested for luciferase detection.

## Plasmid preparation and cell free-expression

Proteins were genetically encoded with enhanced GFP (GFP), mCherry and cMyc (myc) tags, and cloned into the following cell free expression Gateway destination vectors: N-terminal GFP tagged (pCellFree_G03), N-terminal Cherry-cMyc (pCellFree_G07) and C-terminal Cherry-cMyc tagged (pCellFree_G08) (*Gagoski et al., 2015*). Human SOX18 (BC020780), SOX18.Q161* (Modified as

describe below from BC020780), Open Reading Frames (ORFs) were sourced from the Human ORFeome collection, version 5.1, and the Human Orfeome collaboration OCAA collection (Open Biosystems), as previously described (*Sierecki et al., 2013*) and cloned at UNSW. The entry clones pDONOR223 vectors were exchanged with the ccdB gene in the expression plasmid by LR recombination (Life Technologies, Australia). The full-length human *SOX18* gene was synthesized (IDT DNA, USA) and transferred to pCellFree vectors using Gateway PCR cloning. Translation competent *Leishmania tarentolae* extract (LTE) was prepared as previously described (*Kovtun et al., 2011*; *Mureev et al., 2009*). Protein pairs were co-expressed by adding 30 nM of GFP template plasmid and 60 nM of Cherry template plasmid to LTE and incubating for 3 hr at 27°C.

## Construction of the SOX18.Q161* construct

The SOX18.Q161* construct was made by adding a codon stop (UAA) at the original Glutamine acid in position 161 of the Human SOX18 (BC020780).

The SOX18.Q161 was obtained as a gBlock (IDT), and was exchanged with the ccdB gene in the donor plasmid (pDONOR223) by BP recombination (Life Technologies, Australia), then with the ccdB gene in the expression plasmid (pCellFree_G03 and pCellFree_G08) by LR recombination (Life Technologies, Australia) as described previously.

## ALPHAScreen assay

The assay was performed as previously described (*Sierecki et al., 2013*; *Sierecki et al., 2014*), using the cMyc detection kit and Proxiplate-384 Plus plates (PerkinElmer). The LTE lysate co-expressing the proteins of interest was diluted in buffer A (25 mM HEPES, 50 mM NaCl). For the assay, 12.5 µL (0.4 µg) of Anti-cMyc coated Acceptor Beads in buffer B (25 mM HEPES, 50 mM NaCl, 0.001% NP40, 0.001% casein) were aliquoted into each well. This was followed by the addition of 2 µL of diluted sample, at different concentration, and 2 µL of biotin labeled GFP-Nanotrap in buffer A. The plates were incubated for 45 min at room temperature, then 2 µL (0.4 µg) of Streptavidin coated Donor Beads diluted in buffer A was added, and the plate was incubated in the dark for 45 min at room temperature. The ALPHAScreen signal was measured on an Envision Plate Reader (PerkinElmer), using manufacturer's recommended settings ($\lambda_{exc}$ = 680/30 nm for 0.18 s, $\lambda_{em}$ = 570/100 nm after 37 ms). The resulting bell-shaped curve is an indication of a positive interaction, while a flat line reflects a lack of interaction between the proteins. Measurements of each protein pair were performed in triplicate. A binding index was calculated as: $BI = \left[\frac{I - I_{neg}}{I_{ref} - I_{neg}}\right] \times 100$, where $I$ is the highest signal level (top of the hook effect curve) and $I_{neg}$ is the lowest (background) signal level. The signals were normalized to the $I_{ref}$ signal obtained for the strongest interaction.

## Whole mount ear skin staining and cornea staining

Ear skin samples and corneal tissues were fixed in 4% paraformaldehyde (PFA) at 4°C overnight, and immunofluorescence was performed as follows: 1) Blocking solution incubated for 16 hr (10% sheep serum in PBSTx and 1% DMSO); 2) Primary and secondary antibodies were applied overnight at 4°C on a rocker. Wash solution (PBSTx and 1% DMSO) was applied for a few hours at room temperature after each antibody incubation. Ear sample are flat mounted onto a glass slide.

## Quantitation of lymphatic vasculature

After immunofluorescence using anti-Podoplanin (Angiobio anti-mouse 11033), anti-CD31 (Beckton Dickinson anti-rat 550274), ERG (AbCam EPR3864, ab92513) and Endomucin (Santa Cruz Biotechnology sc53941 V.5C7 anti-mouse) antibodies, apical photographs of mouse ears were taken at 20x magnification with a confocal microscope. The averaged field counts for each parameter were collated for all sections before graphical comparisons between phenotypes were generated, and their statistical significance determined by student's paired T-test.

## Immunostaining infantile hemangioma sections

Histological sections from paraffin embedded proliferating infantile hemangiomas were stained with anti-SOX18, anti-CD31, anti-Ki67, anti-D2-40 antibody and nuclei stained with DAPI. Images were captured using a Vectra three multi-spectral imager (Vectra 3.0 Automated Quantitative Pathology

Imaging System, Perkin Elmer) taking advantage of the auto-expose feature of the microscope. Image magnification is 20X objective.

## Acknowledgements

We thank Kristin Johnson for the contribution to figure making and illustration. We thank Dr Mikkio Masuzawa (Kitasato University) for kind gift of the ISO-HAS cell line.

## Additional information

### Funding

| Funder | Grant reference number | Author |
|---|---|---|
| National Health and Medical Research Council | APP 1111169 | Mathias Francois |
| National Heart, Lung, and Blood Institute | R01 HL096384 | Joyce Bischoff |
| National Health and Medical Research Council | APP1107643 | Mathias Francois |

The funders had no role in study design, data collection and interpretation, or the decision to submit the work for publication.

### Author contributions

Jeroen Overman, Conceptualization, Formal analysis, Supervision, Funding acquisition, Investigation, Writing—original draft, Project administration, Writing—review and editing; Frank Fontaine, Conceptualization, Formal analysis, Investigation, Methodology, Writing—review and editing; Jill Wylie-Sears, Conceptualization, Formal analysis, Investigation, Methodology; Mehdi Moustaqil, Formal analysis, Investigation, Methodology; Lan Huang, Marie Meurer, Formal analysis, Investigation; Ivy Kim Chiang, Emmanuelle Lesieur, Formal analysis, Visualization; Jatin Patel, Eddy Pasquier, Formal analysis, Investigation, Writing—original draft; Johannes Zuegg, Resources, Formal analysis, Investigation; Emma Sierecki, Resources, Formal analysis, Supervision, Methodology; Yann Gambin, Resources, Supervision, Methodology; Mohamed Hamdan, Resources, Supervision, Investigation, Methodology, Writing—original draft, Project administration; Kiarash Khosrotehrani, Supervision, Funding acquisition, Investigation, Writing—original draft, Project administration; Gregor Andelfinger, Joyce Bischoff, Mathias Francois, Conceptualization, Formal analysis, Supervision, Funding acquisition, Writing—original draft, Project administration, Writing—review and editing

### Author ORCIDs

Yann Gambin (ID) http://orcid.org/0000-0001-7378-8976
Joyce Bischoff (ID) https://orcid.org/0000-0002-6367-1974
Mathias Francois (ID) https://orcid.org/0000-0002-9846-6882

### Ethics

Human subjects: Hemangioma cell isolation and culture Specimens of infantile hemangioma were obtained under an IRB-protocol approved by the Committee on Clinical Investigation, Boston Children's Hospital. The clinical diagnosis was confirmed in the Department of Pathology. Informed consent was obtained for the specimens, according to the Declaration of Helsinki.

Animal experimentation: All procedures involving mice were approved by the University of Queensland Animal Ethics Committee. All of the animals were handled according to approved institutional animal care and use committee (AEC UQ) protocols (#IMB/049/13/CCQ/NHMRC/CARIPLO/ARC) of the University of Queensland.

### Decision letter and Author response

Decision letter https://doi.org/10.7554/eLife.43026.016
Author response https://doi.org/10.7554/eLife.43026.017

## Additional files

### Supplementary files
• Transparent reporting form
DOI: https://doi.org/10.7554/eLife.43026.014

### Data availability
All data generated or analysed during this study are included in the manuscript and supporting files.

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
