## [Decision Letter]

Thank you for submitting your article "Propranolol in HLTRS and infantile hemangioma: Stereoselective blockage of the SOX18 transcription factor activity" for consideration by *eLife*. Your article has been reviewed by three peer reviewers, including Gou Young Koh as the Reviewing Editor and Reviewer #1, and the evaluation has been overseen by Sean Morrison as the Senior Editor. The following individual involved in review of your submission has agreed to reveal their identity:); Pia Ostergaard (Reviewer #2).

The reviewers have discussed the reviews with one another and the Reviewing Editor has drafted this decision to help you prepare a revised submission.

Summary:

While reviewers 1 and 2 are positive, the reviewers have raised some issues, which are constructive and insightful.

Essential revisions:

We encourage the authors to do additional experiments to address the comments raised by reviewer 1 (comments 4, 5 and 7) and reviewer 3 (comment 2). The remaining comments could be addressed by changes to the text, such as through further discussion.

Reviewer #1:

This study by Overman et al. suggests a new mechanistic insight of propranolol in that R(+)-propranolol enantiomer acts as a small molecule inhibitor of SOX18. They also demonstrate the therapeutic potential of the R(+)-propranolol enantiomer in a patient-derived in vitro model of infantile hemangioma. The hypothesis tested in the study is intriguing and might be clinically relevant with potential therapeutic implications for pathologies involving SOX18 mutations and/or aberrant vascular growth, such as infantile hemangioma. Therefore, I would like to support the publication of this work after addressing the following points.

1) The authors suggest that the R(+)-propranolol enantiomer directly inhibits the transcriptional activity of SOX18 and thereby functions as an anti-angiogenic agent. However, the authors also demonstrate that propranolol restores SOX18 functionality in a dose-dependent manner, which is demonstrated by Vcam1 (SOX18 target) promoter luciferase assay (Figure 2B). This particular aspect seems counterintuitive. Considering that SOX18 mutations in hypotrichosis-lymphedema-telangiectasia and renal syndrome (HLTRS) act in a dominant-negative manner by impairing SOX18 function, please clearly address whether and how the R(+)-propranolol enantiomer restores or inhibits SOX18 function.

2) Could the authors check the protein expression of SOX18 in infantile hemangioma tissue samples or in cell populations derived from infantile hemangioma before and/or after treatment with propranolol to test its beneficial effect in these circumstances?

3) The authors demonstrated that the propranolol treatment in a patient with HLTRS dramatically improved pericardial effusion (Figure 1A-B). This change might have resulted secondarily from the process caused by SOX18 mutation (possibly including lymphedema, and/or renal syndrome), as the primary pericardial effusion itself is a relatively rare disease entity (Imazio et al., Nature Review Cardiology, 2009). Does the HLTRS patient have improvements in symptoms other than pericardial effusion, which might be directly related to HLTRS (e.g. telangiectasia, lymphedema) after propranolol treatment rather than pericardial effusion alone? Furthermore, if the mechanism of action of propranolol in the patient with SOX18 mutation is independent of anti-β-adrenergic activity as the authors suggested, how was the heart function of the patient, including tamponade physiology, improved? Is there any direct or indirect evidence to suggest that SOX18 function is restored or at least altered with propranolol treatment in this patient? These issues might need to be discussed in the Discussion section.

4) The authors suggest that propranolol inhibits SOX18 activity during VEGF-B-induced differentiation of hemangioma-derived stem cells (HemSC) into the hemangioma endothelial cells (HemECs). Could the authors verify that there is no change in SOX18 activity with β-adrenergic agonist treatment to support their claim that the effect of propranolol on SOX18 is independent of a β-adrenergic mechanism?

5) Does propranolol rescue only the aberrant vessel growth in Sox18 *RaOp* mouse cornea? Or also in other tissues such as skin, as the hypotrichosis and infantile hemangioma usually involves and grows on the skin surface.

6) In Figure 2—figure supplement 1B, the authors demonstrate the lymphatic vessel outgrowth (Neuropillin-2 and Prox1) in corneal whole-mount staining of Sox18 *RaOp* mice. This data is not mentioned in the main text although it is demonstrated in the supplemental figure and its legend. Why do the cornea lymphatic vessels outgrow in Sox18 *RaOp* mice?

7) Does the R(+)-propranolol enantiomer modulate VEGFR2 levels or its phosphorylation upon VEGF-A stimulation? This should be easily testable as the authors already have the cell lines, and this might partly explain and support the regression of infantile hemangioma by propranolol treatment.

Reviewer #2:

I have read the manuscript from Overman et al. with great interest. It is very interesting to get an insight into how some of these genetic defects can possibly be overcome using various forms of drugs. Overall, I think this manuscript represents a solid piece of work and I do not have any major comments except that it would be nice if the discussion had been a bit more comprehensive. For example, the authors mention that the propranolol increased the mean weight of the wild type mice, but not of the *RaOp* mice. Why is that?

Reviewer #3:

The key message of this study is that the block of Sox18 is a main 'mechanism of action' of propranolol in treating HLTRS and infantile hemangioma. Several experiments support this conclusion, but insufficiently.

1) Abnormal SOX18 protein derived from mutation in a patient can affect downstream broadly. Thus, it is difficult to judge targets of propranolol in a HLTRS patient. The *RaOp* mouse model has same issue, too.

2) Although both HLTRS and RaOp have mutations in Sox18, phenotypes focused in this study are different: pericardial edema in HLTRS and corneal angiogenesis in RaOp. They seem to be far away for comparison. In addition, the dose of propranolol for a HLTLS patient was 4 mg/kg/day while that for *RaOp* mice was 25 mg/kg/day (with an increasing potential of non-selective effects).

3) Use of the R(+) enantiomer of propranolol was suggested to preclude β-adrenergic-dependent blocking activity. However, the R(+) enantiomer was not used in the assessment of corneal neo-vascularisation.

4) Similarly, differentiation of hemangioma stem cells into endothelial cells relies on many factors including all SoxF members but not Sox18 alone. Downregulated genes by propranolol are representative endothelial markers rather than Sox18 targets (Figure 3C). Altogether, there may be a possibility that propranolol plays a role in a Sox18-independent manner. Additional experiments using other non-selective β-adrenergic blockers such as carvedilol may support the conclusions.

5) Some in vitro experiments appear to lack a physiologic link to in vivo and patient data.

– There was no link to Sox18 function in cell assays shown in Figures 1C and ID.

– The cell systems used in Figures 2B and 2C may be different from lymphatic or hemangioma cells.

– In the present manuscript, only numbers are presented without additional help on ALPHAScreen assay and it is very difficult to figure out how reliable the data of protein-protein interactions in the bar graph are.

– Sox18 protein increased luciferase activity by two-fold in the assay in Figure 2B. This extent of enhanced activity, in general, is not enough to prove binding of transcription factors to their target regions, raising a concern whether the Vcam-1 promoter region used in Figure 2B is a convincing target region of Sox18 in this context.

---

## [Author Response]

Essential revisions:We encourage the authors to do additional experiments to address the comments raised by reviewer 1 (comments 4, 5 and 7) and reviewer 3 (comment 2). The remaining comments could be addressed by changes to the text, such as through further discussion.Reviewer #1:This study by Overman et al. suggests a new mechanistic insight of propranolol in that R(+)-propranolol enantiomer acts as a small molecule inhibitor of SOX18. They also demonstrate the therapeutic potential of the R(+)-propranolol enantiomer in a patient-derived in vitro model of infantile hemangioma. The hypothesis tested in the study is intriguing and might be clinically relevant with potential therapeutic implications for pathologies involving SOX18 mutations and/or aberrant vascular growth, such as infantile hemangioma. Therefore, I would like to support the publication of this work after addressing the following points.1) The authors suggest that the R(+)-propranolol enantiomer directly inhibits the transcriptional activity of SOX18 and thereby functions as an anti-angiogenic agent. However, the authors also demonstrate that propranolol restores SOX18 functionality in a dose-dependent manner, which is demonstrated by Vcam1 (SOX18 target) promoter luciferase assay (Figure 2B). This particular aspect seems counterintuitive. Considering that SOX18 mutations in hypotrichosis-lymphedema-telangiectasia and renal syndrome (HLTRS) act in a dominant-negative manner by impairing SOX18 function, please clearly address whether and how the R(+)-propranolol enantiomer restores or inhibits SOX18 function.

In this study we show that propranolol has the ability to disrupt both SOX18 protein-protein interactions and SOX18-SOX18mut (RaOp) protein-protein interactions (i.e. SOX18 homodimer and SOX18/SOX18RaOp heterodimer formation Figure 2C). In the context of a transcription-based assay (Figure 2B) SOX18 *RaOp* mutant protein is a potent repressor of SOX18 activity (even up to a 30wt/1RaOp ratio), this causes the loss of Vcam1 transactivation (Figure 2B). SOX18 *RaOp* mutant protein acts by “poisoning functional SOX18 protein complexes” by forming SOX18-RaOp dimers. SOX18 function is partly restored by propranolol because propranolol disrupts SOX18RaOP protein recruitment and interaction with SOX18 WT. The non “poisoned” SOX18 is then able to transactivate Vcam1.

In other words the process corresponds to the inhibition (propranolol) of an inhibitor (RaOp) which in turn results to the restoration of SOX18 active form. Please find in Author response image 1 a schematic of the proposed partial agonist mechanism.

**Author response image 1. respfig1:** Propranolol rescues the RaOp protein-mediated inhibition by disrupting the interaction SOX18/SOX18RaOp.

2) Could the authors check the protein expression of SOX18 in infantile hemangioma tissue samples or in cell populations derived from infantile hemangioma before and/or after treatment with propranolol to test its beneficial effect in these circumstances?

We have assessed SOX18 expression in 3 different human infantile hemangioma tissue samples from patients that did not received propranolol treatment, as patients treated with propranolol do not typically undergo surgery. We detected expression of the SOX18 protein in the pathological blood vasculature by immunofluorescence. The SOX18 protein was detected in a large subset of CD31-positive endothelial cells, in accordance with its role during hemangioma stem cell to EC differentiation process. This new result is now incorporated into Figure 3 as Figure 3A and reported in the text (Results section).

We next examined the effect of Sm4 and propranolol on SOX18 gene expression levels in the differentiated HemSCs (n=4). As shown in Author response image 2 Panel A, SOX18 increased upon VEGF-B-induced endothelial differentiation (p=0.0026), consistent with results shown in Figure 3B; inclusion of Sm4 or propranolol during the differentiation assay decreased levels of SOX18 transcript, measured by qPCR. This result is expected because both inhibitors block HemSC-EC differentiation (Figure 3C and Figure 3—figure supplement 2) and therefore decrease the total number of mature ECs. We normalised the data relative to a pan-endothelial cell marker such as CD31 (Panel B), to account for the bias introduced by the lack of endothelial cells differentiation. Using CD31 as a reference gene we show that SOX18 levels were constant, with a small but significant increase when propranolol was included in the differentiation media.

**Author response image 2. respfig2:** HemSC were treated without (control) or with VEGF-B for 5 days to induce endothelial differentiation. Sm4 or propranolol were included in the differentiation medium at 5μM. DMSO, the vehicle, was added at the same dilution to serve as control. SOX18 was measured by qPCR and normalized to the house keeping transcript ATP5B (**A**) or to the pan-endothelial marker CD31 (**B**). Data points are from 4 different experiments carried out with HemSC from three different infantile hemangiomas.

3) The authors demonstrated that the propranolol treatment in a patient with HLTRS dramatically improved pericardial effusion (Figure 1A-B). This change might have resulted secondarily from the process caused by SOX18 mutation (possibly including lymphedema, and/or renal syndrome), as the primary pericardial effusion itself is a relatively rare disease entity (Imazio et al., Nature Review Cardiology, 2009). Does the HLTRS patient have improvements in symptoms other than pericardial effusion, which might be directly related to HLTRS (e.g. telangiectasia, lymphedema) after propranolol treatment rather than pericardial effusion alone?

The patient had mild improvement in both the telangectasia (especially face and lips) and lymphedema (evidenced by reducing the dose of the diuretic Furosemide). However, the dramatic improvement was in the complete resolution of the pericardial effusion which was sustained even after stopping the propranolol. The exposure to propranolol was rather short, only 16 months of treatment were necessary to improve the pericardial effusion.

It is important to bear in mind that most of HLTRS symptoms are acquired during embryonic development when SOX18 is at play during the molecular process of endothelial cell specification and hair stem cell differentiation. It is therefore not expected to correct symptoms acquired during early development especially when the molecular target is not at play post-natally.

Furthermore, if the mechanism of action of propranolol in the patient with SOX18 mutation is independent of anti-β-adrenergic activity as the authors suggested, how was the heart function of the patient, including tamponade physiology, improved?

The systolic cardiac function (ejection fraction) was normal even during the time when there was large pericardial effusion. The diastolic function however was decreased due to the effusion. Nevertheless, the diastolic dysfunction observed did not reach the degree of tamponade.

Is there any direct or indirect evidence to suggest that SOX18 function is restored or at least altered with propranolol treatment in this patient? These issues might need to be discussed in the Discussion section.

So far the only indirect evidence that SOX18 function is restored has been reported in Wunnemann et al., 2015 whereby after years of propranolol treatment (from a young age to 13yo) the HTLRS patient did not develop alopecia and display normal hair growth.

We agree with the comment. With the current amount of scientific evidence we cannot conclude that in HLTRS the effects are mediated via a partial restoration of SOX18 function only. This could be due to an additive or synergistic effect with β-adrenergic blockade. This is now addressed in the Discussion.

4) The authors suggest that propranolol inhibits SOX18 activity during VEGF-B-induced differentiation of hemangioma-derived stem cells (HemSC) into the hemangioma endothelial cells (HemECs). Could the authors verify that there is no change in SOX18 activity with β-adrenergic agonist treatment to support their claim that the effect of propranolol on SOX18 is independent of a β-adrenergic mechanism?

We tested whether or not inclusion of isoproterenol, a non-selective β-adrenergic receptor agonist, during the VEGF-B-induced differentiation would alter SOX18 activity or increase endothelial differentiation. We measured NOTCH1, a SOX18 target gene, and VE-cadherin, an EC marker, and SOX18 in VEGF-B treated HemSCs cells by qPCR. In four biological replicates with HemSC isolated from three different infantile hemangioma specimens, we did not detect a significant increase in VE-cadherin, NOTCH1 or SOX18 in isoproterenol-treated cells compared to DMSO (vehicle)-treated cells. Propranolol caused a significant decreased in VE-cadherin, NOTCH1 and SOX18 transcripts; p values <0.05 (consistent with Figure 3C and Figure 3—figure supplement 2.) This suggests SOX18 activity is not modulated by a β-adrenergic pathway. Isoproterenol in the absence of VEGF-B-induced differentiation had no effect on VE-cadherin, NOTCH1, SOX18 levels (Author response image 3).

**Author response image 3. respfig3:** HemSC were treated with VEGF-B for 5 days to induce endothelial differentiation. Propranolol or isoproterenol were included in the differentiation medium at 5μM, whereas DMSO, the vehicle, was added at the same dilution to serve as control. VE-cadherin, NOTCH1 and SOX18 were measured by qPCR. Data points are from 4 different experiments carried out with HemSC from three different infantile hemangiomas.

5) Does propranolol rescue only the aberrant vessel growth in Sox18 RaOp mouse cornea? Or also in other tissues such as skin, as the hypotrichosis and infantile hemangioma usually involves and grows on the skin surface.

We have performed an additional in vivo experiment to assess whether propranolol, the R-enantiomer, or the S-enantiomer impact aberrant vessel patterning in skin tissues. Animals were treated at 25mg/Kg/day (oral gavage) from P1 to P28. Ear skin tissues were collected and whole mount immunofluorescence staining was performed for CD31, Podoplanin and ERG. Samples were imaged by confocal microscopy and vessel morphogenesis was quantified using IMARIS software.

As previously published, we observed that the *Ragged Opossum* heterozygous mutant mice display severe lymphatic vascular defects (Francois et al., 2008) with major morphogenesis defects of the dermal lymphatic vasculature. The loss of a significant portion of vessels with valve structure (pre-collectors and collectors) revealed a dramatic phenotype of the mature lymphatic vascular network in the *RaOp* animals (Figure 3—figure supplement 4, top graph).

There was no phenotypic rescue corresponding to a significant improvement of the lymphatic vasculature mis-patterning observed for any of the drug treatments in the mutant animals. This result is in accordance with the fact the drug treatment occurs at post-natal stages whereas the dermal lymphatic vascular defects are acquired from early to mid-stages of organogenesis in the embryo. This result is now indicated in the manuscript.

For transparency of the review process we have appended in Author response image 4 all the images we have used for quantification of the skin lymphatic vascular experiment.

**Author response image 4. respfig4:** Images of whole mount skin immuno-stained for podoplanin. Each image corresponds to the same region of interest from the inner skin flap of a mouse ear. Untreated WT, n=4; untreated ragged n=3; S(-) propranolol WT n=4; S(-) propranolol ragged n=3; propranolol WT n=5, propranolol ragged n=4; R(+) propranolol WT n=6, R(+) propranolol ragged n=4.

6) In Figure 2—figure supplement 1B, the authors demonstrate the lymphatic vessel outgrowth (Neuropillin-2 and Prox1) in corneal whole-mount staining of Sox18 RaOp mice. This data is not mentioned in the main text although it is demonstrated in the supplemental figure and its legend. Why do the cornea lymphatic vessels outgrow in Sox18 RaOp mice?

The molecular mechanism that underpins vascular outgrowth in the cornea for both blood and lymphatic vessels remains unclear. Since most causes of corneal neo-vascularisation (CNV) are indeed VEGF-dependent, and some are successfully treated with anti-VEGF based therapies, we decided to focus on the potential contribution of augmented VEGF signalling to the invasion of blood vessel into the *Sox18^RaOp^* corneas. In the context of CNV, *Flt1* is of particular interest, as the balance of distinct isoforms of this protein has been shown to be of key importance in maintaining corneal avascularity (Ambati et al., 2006). The pro- or anti-angiogenic activity of FLT1 is highly dependent on the particular splice variant that is produced (Kearney et al., 2004, Ferrara et al., 2003, Claesson-Welsh, 2016). The shorter isoform sFLT is released into the extracellular matrix to sequester endogenous pro-angiogenic VEGF-A, preventing the activation of FLT1 and VEGFR2 receptors and any downstream cellular response. Deficiency of sFLT in the cornea has been shown to be causative for CNV (Ambati et al., 2006). Further, we have recently shown that FLT1 is a potential direct target gene of SOX18 dimer complex (Moustaqil et al., NAR 2018). This observation prompted us to investigate FLT1 expression levels in the *RaOp* mutant mouse.

In order to identify whether VEGF pathway is dysregulated in the tissue of the mutant animals we have performed qPCR analysis for VEGF receptors and soluble receptors on the whole cornea of wild type and mutant animals at different time points (8weeks, P12weeks, P16weeks). See Author response image 5. To distinguish between the different splice variants that give rise to either the full length FLT1 receptors or soluble FLT1 (sFLT1), we designed primers to specifically amplify the following regions:

‘*Flt1 (extra)’*, extracellular domain that is present in both full length FLT1 and sFLT1;

‘*Flt1 (intra)’*, intracellular domain that is only present in full length FLT1;

‘*sFlt1’*, unique 3’end of exon 12 and 3’UTR only present on the *sFlt1* transcript

**Author response image 5. respfig5:** Whole corneas were harvested from 8, 12 and 16 week old mice and prepared for qRT-PCR analysis. Gene transcripts for *Pecam, Flt1, Vegfr2* and *Foxc* were normalized to either housekeeper gene Rpl13 (top row) or vascular gene Pecam (bottom row) to normalize for total amount of endothelial cells (EC). *SOX18^RaOp^* mice at 8 weeks of age had increased levels of *Pecam*, indicating an increase in ECs, and increased levels of the transcripts corresponding to the extracellular domain of the Flt1 receptor (*Flt1*-extra), the intracellular domain (*Flt1*-inra) and the exclusively soluble variant of *Flt1 (sFlt1*). * P-value ≤ 0.05, Holm-Sidak multiple comparison test. Individual biological replicates are shown, including mean ± s.e.m of n=2-8..

The most obvious changes in *Flt1* levels were observed at 8 weeks, in agreement with the expression pattern of *Pecam* (Figure 2—figure supplement 1C). At this stage, the amount of total *Flt1 (Flt1* extra) in the *Sox18^RaOp^* corneas was 4.6 fold higher than in the wild type (*p*-value < 0.05). This is partly explained by the fact that these corneas are covered in blood vessels, and the increased amount of endothelial cells contributes to the induction of *Flt1* over the whole tissue. However, the increase in *Pecam* transcript levels – representing the increase in endothelial cells – is much lower than that of *Flt* (2 fold). In order to compensate for the increase in endothelial cells, we normalized the *Flt1* transcript levels to *Pecam* levels, which makes it evident that total *Flt1* increased disproportionality to the amount endothelial cells (2.0 fold, *p*-value < 0.05).

Although total *Flt1* is upregulated in the corneas of *SOX18^RaOp^* mice, *sFlt1* levels are unchanged regardless of the amount of vascular endothelial cells. This suggests that the ratio between the amounts of full length FLT1 and sFLT1 in the cornea shifted in favour of the full-length isoform. Transcript levels for *Vegfr2* were not significantly different between wild types and mutants at 8 weeks. Since FLT1, VEGFR2 and sFLT1 all compete for the same pool of VEGF-A, the shift in *Flt1* isoforms could explain the corneal phenotype. However, whether exactly this is causative for the penetration of vessels into the cornea of *Sox18^RaOp^* mice, and whether SOX18 is directly involved in regulating the balance between *Flt1/sFlt1*, is uncertain at this stage.

Of note FOXC1 levels remains unchanged; we tested for this gene expression level since loss of this transcription factor function has been shown to cause corneal neo-vascularisation (Axenfeld-Rieger Syndrome) (Mears et al. Am j Hum Genet 1998)

7) Does the R(+)-propranolol enantiomer modulate VEGFR2 levels or its phosphorylation upon VEGF-A stimulation? This should be easily testable as the authors already have the cell lines, and this might partly explain and support the regression of infantile hemangioma by propranolol treatment.

We address this question with three experiments.

In panel A, normal human ECFCs were pre-treated for 1 hour with propranolol, R+ enantiomer or S-enantiomer (each tested at 5μM) and then stimulated for 5 minutes with VEGF-A. Cells were lysed and analysed for phosphorylated VEGFR2 and total VEGFR2 by Western blotting. The drug treatments had no effect on levels of VEGFR2 or levels of VEGF-A-stimulated phosphorylation. The same results were seen when drug pre-treatment was carried out for 16 hours (not shown).

In Panel B, HemSC were induced to undergo endothelial differentiation with VEGF-B for 5 days, then pre-treated with propranolol, R+ enantiomer or S-enantiomer for 1 hour and next stimulated with VEGF-A for 5 minutes. Western blotting for VEGFR2 shows the drug treatments had no effect on VEGFR2 levels; the same results were seen with HemSC isolated from two different IH.

In Panel C, the same experiment was performed and VEGFR2 was immunoprecipitated to increase detection of phosphorylated VEGFR2. Drug pre-treatment had no effect on phosphor-VEGFR2 levels. The lack of VEGF-A-induced phosphorylation of VEGFR2 is consistent with previous reports wherein VEGFR2 is primarily intracellular in HemSC (Khan et al., 2008) and is constitutively phosphorylated in HemEC (Jinnin et al., Nat Med 2008). In summary, we did not observe any changes to VEGFR2 protein levels or its phosphorylation upon R-enantiomer, propranolol or S-enantiomer treatment. This is now shown in the manuscript as Figure 3—figure supplement 2B-D.

Reviewer #2:I have read the manuscript from Overman et al. with great interest. It is very interesting to get an insight into how some of these genetic defects can possibly be overcome using various forms of drugs. Overall, I think this manuscript represents a solid piece of work and I do not have any major comments except that it would be nice if the discussion had been a bit more comprehensive. For example, the authors mention that the propranolol increased the mean weight of the wild type mice, but not of the RaOp mice. Why is that?

One of the phenotypes of the *Ragged Opossum* mutant is a complete lack of sub-cutaneous fat (Francois et al unpublished). The absence of adipose tissue makes it impossible for adipocytes to respond to a beta-adrenergic treatment.

It is for this particular reason that we think the *Ragged Opossum* is not gaining weight under propranolol treatment. The adipose tissue phenotype, currently under investigation, is beyond the scope of this study.

Reviewer #3:The key message of this study is that the block of Sox18 is a main 'mechanism of action' of propranolol in treating HLTRS and infantile hemangioma. Several experiments support this conclusion, but insufficiently.1) Abnormal SOX18 protein derived from mutation in a patient can affect downstream broadly. Thus, it is difficult to judge targets of propranolol in a HLTRS patient. The RaOp mouse model has same issue, too.

This concern is directed to the in vivo data analysis of propranolol treatment. On the other hand, our data from in vitro and cellular assays shows strongly that propranolol mode of action is mediated via a SOX18-dependent mechanism.

The recent description of the molecular function of the SOX18 homodimer in the context of endothelial cell differentiation (Moustaqil et al. NAR 2018) is an important piece of information. Here we show that the *RaOp* mutant protein has the capability to recruit SOX18 wild type protein and therefore interfere with the endothelial specific signature of SOX18 wild type homodimer.

Further, in this study we show that use of specific β adrenergic blockers or loss of β-adrenergic receptors does not affect ECFC or cancer cell line survival. As a final demonstration of a role of SOX18-mediated mode of action in vivo we now show that treatment with the R-enantiomer of propranolol (low to no anti-β adrenergic activity) is able to rescue the corneal neo-vascularisation phenotype.

2) Although both HLTRS and RaOp have mutations in Sox18, phenotypes focused in this study are different: pericardial edema in HLTRS and corneal angiogenesis in RaOp. They seem to be far away for comparison. In addition, the dose of propranolol for a HLTLS patient was 4 mg/kg/day while that for RaOp mice was 25 mg/kg/day (with an increasing potential of non-selective effects).

The point of using the *Ragged Opossum* mouse model experiment is to show on target engagement by rescuing a phenotype which relies on the activity of the SOX18 mutant protein. We agree that the readout in human is different, but in this particular case the decision to pharmacologically manage a new HLTRS case with propranolol was guided by the results obtained from pre-clinical data. This provided a therapeutic option where none was available before.

The choice of the dosage was originally guided by our experience in vivo in mice using Sm4 (SOX18 small molecule inhibitor) which we have shown is a potent anti-metastatic compound at 25mg/kg/day in pre-clinical model of breast cancer (Overman et al eLife 2017).

Of note we have shown efficacy of Sm4 at lower concentration in a dose dependent manner (from 5 to 50mg/Kg/day, unpublished data). This suggests that the patient treated with low dose of propranolol (4mg/Kg/day) has received a dose of small molecule inhibitor that is able to block SOX18 activity.

In the *Ragged Opossum* in vivo experiment we aimed at a maximum tolerated dose to get the best rescuing effect possible in turn translating into an obvious phenotypic rescue. The concentration scouting is a great suggestion but is not necessary as all in vitro data show concentration dependence and that the main goal of the rescuing approach is to show selective on-target engagement.

3) Use of the R(+) enantiomer of propranolol was suggested to preclude β-adrenergic-dependent blocking activity. However, the R(+) enantiomer was not used in the assessment of corneal neo-vascularisation.

We have now assessed the effect of the R(+) enantiomer in vivo in a rescue experiment of the corneal neo-vascularisation phenotype in the *Ragged Opossum (RaOp*) mouse model. These results are now included in the manuscript as a supplemental figure to Figure 3.

In short, the R(+) enantiomer has the ability to rescue the corneal neo-vascularisation phenotype of the *RaOp* heterozygous mouse. Further, we have also tested the S(-) enantiomer in the same model system. Interestingly it seems that this compound has the potential to rescue the phenotype albeit to a lesser extent. This results suggests that both R(+) and S(-) enantiomers have the capability to inhibit the SOX18 *RaOp* mutant protein activity. This is most likely by disturbing the SOX18/SOX18RaOp or the RBPJ/SOX18RaOp protein-protein interactions as shown by our in vitro assays. Of importance, in the hemangioma hemSC-EC differentiation assay the R(+) enantiomer was as efficient as propranolol in halting this differentiation process, whereas the S(-) enantiomer showed a weak inhibitory response, suggesting an enantiomer selective blockade of the SOX18 wild type protein mediated by the R(+) enantiomer.

Because of the weak activity of the S(-)enantiomer, we have revised the title of our manuscript to reflect accurately the key finding of this study: *“R-propranolol is a small molecule inhibitor of the SOX18 transcription factor in a rare vascular syndrome and hemangioma.”*

4) Similarly, differentiation of hemangioma stem cells into endothelial cells relies on many factors including all SoxF members but not Sox18 alone. Downregulated genes by propranolol are representative endothelial markers rather than Sox18 targets (Figure 3C). Altogether, there may be a possibility that propranolol plays a role in a Sox18-independent manner. Additional experiments using other non-selective β-adrenergic blockers such as carvedilol may support the conclusions.

We did not test additional non-selective β-adrenergic blockers.

5) Some in vitro experiments appear to lack a physiologic link to in vivo and patient data.– There was no link to Sox18 function in cell assays shown in Figures 1C and ID.

The point of experiments shown in Figure 1C and 1D is to show that propranolol acts has a mode of action which is at least in part independent to β-adrenergic receptors. The link of propranolol to SOX18 is provided by other assay such as the in vitro transcriptional report assay (Figure 2B) and ALPHAScreen assay (Figure 2C).

– The cell systems used in Figures 2B and 2C may be different from lymphatic or hemangioma cells.

These in vitro assays are synthetic and designed to show propranolol on target engagement on SOX18 activity.

We actually use a cell based assay where SOX18 is not expressed and add exogenous SOX18 to demonstrate specific inhibition of this TF by propranolol.

– In the present manuscript, only numbers are presented without additional help on ALPHAScreen assay and it is very difficult to figure out how reliable the data of protein-protein interactions in the bar graph are.

We have now provided more information on the statistical analysis of the ALPHAScreen data analysis in the text of the corresponding figure legend.

– Sox18 protein increased luciferase activity by two-fold in the assay in Figure 2B. This extent of enhanced activity, in general, is not enough to prove binding of transcription factors to their target regions, raising a concern whether the Vcam-1 promoter region used in Figure 2B is a convincing target region of Sox18 in this context.

Although we agree that this assay efficacy window is not optimum, this result is highly reproducible and quite robust. The transactivation of the Vcam1 promoter fragment and the regulation of Vcam1 by SOX18 has been previously characterised and published, please refer to Hosking et al., 2004.